# DUAL-PATH CONDITION ALIGNMENT FOR DIFFUSION TRANSFORMERS

**Changhao Peng[1], Yuqi Ye[1], Shuangjun Du[2], Wenxu Gao[1], Wei Gao[1]***

[1]School of Electronic and Computer Engineering, Peking University
[2]Institute of Automation, Chinese Academy of Sciences
*Corresponding author.

## ABSTRACT

Denoising-based generative models have been significantly advanced by representation-alignment (REPA) loss, which leverages pre-trained visual encoders to guide intermediate network features. However, REPA's reliance on external visual encoders introduces two critical challenges: potential *distribution mismatches* between the encoder's training data and the generation target, and the high *computational costs* of pre-training. Inspired by the observation that REPA primarily aids early layers in capturing robust semantics, we propose an unsupervised alternative that avoids external visual encoder and the assumption of consistent data distribution. We introduce **DUal-Path condition Alignment** (**DUPA**), a novel self-alignment framework, which independently noises an image multiple times and processes these noisy latents through decoupled diffusion transformer, then aligns the derived conditions—low-frequency semantic features extracted from each path. Experiments demonstrate that DUPA achieves FID=1.46 on ImageNet 256×256 with only 400 training epochs, outperforming all methods that do not rely on external supervision. DUPA is also model-agnostic and can be readily applied to any denoising-based generative model, showcasing its excellent scalability and generalizability. Code is available at https://github.com/PCH-gg/DUPA, https://openi.pcl.ac.cn/OpenAIDriving/DUPA.

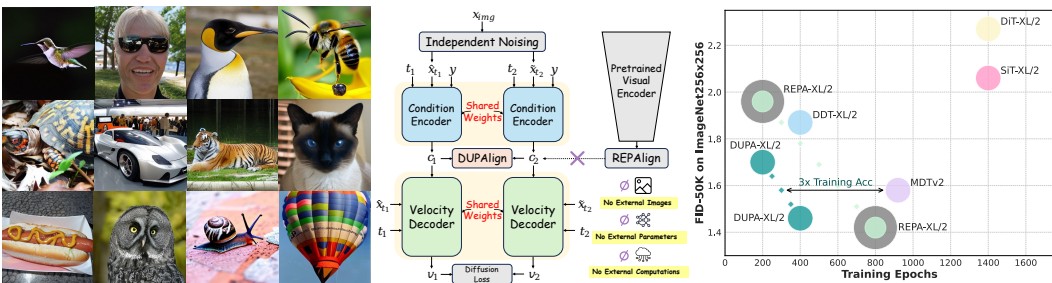

Figure 1: **Unsupervised representation alignment can efficiently train diffusion transformer as REPA does.** By aligning the representations of different noised images, DUPA achieves FID performance comparable to that of REPA with only 400 training epochs, which means $\geq 3\times$ faster convergence than current state-of-the-art methods that do not rely on supervision from an external visual encoder. The radius of the circles in the right figure denotes model size while the gray ring surrounding REPA represents the auxiliary visual encoder.

## 1 INTRODUCTION

In recent years, denoising-based generative models (Peebles & Xie, 2023; Ma et al., 2024) have achieved remarkable progress in modeling complex data distributions. Such models are typically composed of stacking transformer blocks. REPA (Yu et al., 2025) points out that aligning the intermediate representations of transformer blocks with the features extracted by high-performance

visual encoders (e.g., CLIP (Radford et al., 2021), DINOv2 (Oquab et al., 2024), etc.) can significantly enhance the performance of generative models. Since the proposal of REPA, most methods in class-to-image generation tasks have been built upon this approach.

However, applying REPA to specific application scenarios may face the following challenges from our perspective:

**Out of distribution.** If there is a significant discrepancy between the data distribution modeled by the generative model and the pre-training distribution of the large visual encoder, the features extracted by the visual encoder may not only fail to facilitate the training of the generative model but could also potentially "mislead" it, resulting in performance degradation.

**Additional computational costs.** Both pre-training and fine-tuning large visual encoders for specific application scenarios incur additional computational costs. For instance, pre-training DINOv2 requires 1.1 billion model parameters, 1,500 training epochs, and 142 million images—far exceeding the computational resources needed to train DiT (Yao et al., 2024) or SiT(Ma et al., 2024). Moreover, if the data distribution in a specific domain differs from the pre-training distribution, further fine-tuning of the visual encoder is necessary, which further increases the computational costs.

Xie et al. point out in REPA: "*Limiting regularization to the first few layers further enhances generation performance. We hypothesize that this enables the remaining layers to concentrate on capturing high-frequency details, building on a strong representation.*" Similarly, Wang et al. note in Decoupled Diffusion Transformer (Wang et al., 2025): "*Current diffusion transformers are fundamentally constrained by their low-frequency semantic encoding capacity.*" Therefore, we posit that the primary contribution of REPA lies in providing accurate and invariant representations derived from pure images to the first few transformer blocks when they extract semantic features from noisy images. As illustrated on the left of Figure 2, REPA acts like a "*data annotator*" during training, supplying "*labels*" (*i.e.*, effective representations) obtained from "*ground truth*"(*i.e.*, pure images) for noisy images, which is similar to supervised learning. However, as discussed above, this "supervised learning" approach in REPA faces two challenges compared to unsupervised learning: "costliness of labeling" and "inaccurate labeling" issues. Consequently, **we aim to utilize unsupervised learning to provide effective representation guidance for generative model training**, much like REPA does but without the assumption of consistent data distribution and expensive additional computational costs.

Recently, several works have incorporated unsupervised learning into generative model training to improve performance. Broadly, we categorize these works into two types: introducing *masked image modeling* into the denoising process to enhance the contextual reasoning ability of generative models, such as MaskDiT (Zheng et al., 2024) and SD-DiT (Zhu et al., 2024); and utilizing intermediate representations of generative models for *contrastive learning* (typically treating them as negative pairs) to improve training efficiency, such as Contrastive Flow Matching (Stoica et al., 2025) and Dispersive Loss (Wang & He, 2025). However, neither of these unsupervised approaches can provide accurate representation guidance for each image in the way REPA does, making it difficult for their performance to match that of REPA.

Based on the above insights, we propose **DUal-Path condition Alignment** (DUPA). As shown on the right of Figure 2, an image is independently noised multiple times during training, and use Decoupled Diffusion Transformer to predict different denoising paths. In this way, the condition encoder can extract different conditions, which are low-frequency semantic features from different noisy images. Since these conditions originate from the same pure image, they should be similar, much like the representations obtained by large visual encoders in REPA. We propose to align these different conditions derived from independently noised versions of a single image to furnish effective representation guidance for model training. In summary, our contributions can be outlined as follows:

- We point out that REPA may face issues of out of distribution and high computational costs, and hypothesize that internal alignment of noisy images can also provide effective representation guidance for training of diffusion transformer without external supervision.
- We introduce DUPA, a simple alignment for two noisy views of a single image without external supervision, which can be easily applied to other denoising-based generative models.
- Our proposed DUPA achieves a remarkable FID of 1.46 after only 400 training epochs, surpassing all evaluated methods that do not rely on external supervision. It also significantly narrows the

performance gap with REPA (FID=1.42), a model trained for 800 epochs under the guidance of external visual encoders. Furthermore, compared to DUPA's base model, DUPA accelerates training by $5\times$ and inference by $10\times$.

## 2 RELATED WORKS

### 2.1 DIFFUSION TRANSFORMERS WITH REPRESENTATION LEARNING

Diffusion transformers (Peebles & Xie, 2023) present an innovative architecture for diffusion models which integrates transformers (Vaswani et al., 2023) into the diffusion framework, effectively replacing the conventional U-Net structure. Studies demonstrate that this architecture can surpass traditional methods particularly when sufficiently trained. SiT (Ma et al., 2024) further validates the effectiveness of transformers and extends their application to challenging tasks such as text-to-image generation (Chen et al., 2023; 2024). Furthermore, diffusion transformers have achieved remarkable progress in the text-to-video domain, exhibiting outstanding visual and motion quality (Hong et al., 2022; Kong et al., 2025).

### 2.2 REPRESENTATION LEARNING IN DIFFUSION MODELS

In image generation research, REPA leverages auxiliary representation learning to optimize generative models by aligning their intermediate representations with those of high-capacity pretrained encoders trained on external data. Building on this foundation, SARA (Chen et al., 2025) innovates by incorporating structured and adversarial alignment strategies. SoftREPA (Lee et al., 2025) extends this approach to the multimodal domain by aligning noisy image representations with soft semantic embeddings. While these approaches demonstrate strong performance in practice, they exhibit a high dependency on additional pretraining and external data.

### 2.3 UNSUPERVISED LEARNING IN DIFFUSION MODELS

The integration of masked image modeling(Xie et al., 2022) into diffusion transformers significantly enhances training efficiency and semantic representation. By masking image tokens during training, masked image modeling forces the model to learn contextual reasoning within the diffusion process, often using an asymmetric encoder-decoder structure that reduces computational cost. This approach accelerates training, improves generation quality, and enables zero-shot image editing capabilities like inpainting. Models such as MaskDiT (Zheng et al., 2024) and MDTv2(Gao et al., 2023b) demonstrate its effectiveness in producing high-quality images with better structural coherence.

Compared to masked image modeling, contrastive learning (Khosla et al., 2020) has recently been demonstrated to be a simpler yet also effective unsupervised method for improving diffusion transformer training. These methods primarily work by constructing negative samples to separate distinct representations. Contrastive Flow Matching (Stoica et al., 2025) proposes to significantly reduce the number of sampling steps required during inference by maximizing the dissimilarities between the predicted velocity and the ground-truth velocity of an image from another category. Dispersive Loss (Wang & He, 2025) suggests that maximizing pairwise distances among different intermediate representations within the same batch can enhance the generative capability of diffusion transformers without considering whether these representations belong to the same category.

## 3 PRELIMINARIES

### 3.1 FLOW AND DIFFUSION-BASED MODELS

Based on the unified framework of stochastic interpolants, flow and diffusion-based models are characterized by a continuous-time interpolation process between data and noise $\mathbf{x}_t = \alpha_t \mathbf{x}_* + \sigma_t \epsilon$, where $\mathbf{x}_* \sim p(\mathbf{x})$ is data and $\epsilon \sim \mathcal{N}(\mathbf{0}, \mathbf{I})$ is Gaussian noise, with $\alpha_t$ decreasing and $\sigma_t$ increasing in time $t$. The dynamics are governed by a probability flow ODE $\dot{\mathbf{x}}_t = \mathbf{v}(\mathbf{x}_t, t)$, enabling deterministic sampling, and an equivalent reverse SDE

$$d\mathbf{x}_t = \mathbf{v}(\mathbf{x}_t, t)dt - \frac{1}{2}w_t \mathbf{s}(\mathbf{x}_t, t)dt + \sqrt{w_t}d\bar{\mathbf{w}}_t, \tag{1}$$

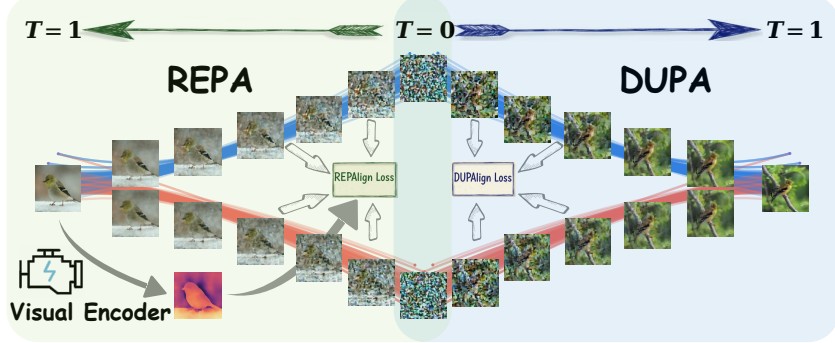

Figure 2: **Comparison between REPA and DUPA.** REPA needs an external visual encoder to generate effective representations, whereas DUPA can get effective representations through internal alignment.

enabling stochastic sampling. The velocity field

$$\mathbf{v}(\mathbf{x}, t) = \dot{\alpha}_t \mathbb{E}[\mathbf{x}_* | \mathbf{x}_t = \mathbf{x}] + \dot{\sigma}_t \mathbb{E}[\epsilon | \mathbf{x}_t = \mathbf{x}] \tag{2}$$

is trained by minimizing the objective

$$\mathcal{L}_{\text{velocity}}(\theta) = \mathbb{E}_{\mathbf{x}_*, \epsilon, t} \left[ \| \mathbf{v}_\theta(\mathbf{x}_t, t) - \dot{\alpha}_t \mathbf{x}_* - \dot{\sigma}_t \epsilon \|^2 \right], \tag{3}$$

unifying both ODE and SDE-based generation approaches.

## 3.2 DECOUPLED DIFFUSION TRANSFORMER

Decoupled Diffusion Transformer (DDT) (Wang et al., 2025) introduces a novel encoder-decoder architecture to resolve the optimization dilemma in traditional diffusion transformers between low-frequency semantic encoding and high-frequency detail decoding.

Specifically, DDT uses a dedicated condition encoder to extract semantic condition features $\mathbf{z}_t = \mathbf{Encoder}(\mathbf{x}_t, t, y)$ and a velocity decoder to predict the velocity field $\mathbf{v}_t = \mathbf{Decoder}(\mathbf{x}_t, t, \mathbf{z}_t)$. This encoder-decoder architecture significantly improves training efficiency while reducing FID (Deng et al., 2009).

## 4 DUPA: DUAL-PATH CONDITION ALIGNMENT

### 4.1 DUAL-PATH SAMPLING

For an input image $\mathbf{x}$ and its class label $y$, we conduct multiple samplings to get different noises $\epsilon_k$ and timestamps $t_k$, thereby generating distinct noisy latents $\mathbf{x}_{t_k} = \alpha_{t_k} \cdot \mathbf{x} + \sigma_{t_k} \cdot \epsilon_k, 1 \le k \le K$ to be denoised, where $K$ represents the number of independent sampling times.

Then we use DDT to estimate the velocity for $\mathbf{x}_{t_k}$:

$$\mathbf{z}_{t_k} = \mathbf{Encoder}(\mathbf{x}_{t_k}, t_k, y), \ \mathbf{v}_{t_k} = \mathbf{Decoder}(\mathbf{x}_{t_k}, t_k, \mathbf{z}_{t_k}). \tag{4}$$

Considering the overall performance and computational cost trade-off (refer to Figure 3a), we set $K = 2$. Multiple independent noise sampling of a single pure image are performed for two main reasons.

**Training efficiency.** It enables the training of different noised states of an image through a single training step. As will be discussed in Section 5.4, this approach is more efficient compared to applying only a single noising operation.

**Different conditions to align.** Multiple independent noise sampling can obtain different velocity conditions for decoding velocities of distinct paths with the same "end point" via DDT. By aligning these conditions, DDT can encode more accurate low-frequency semantic information, which will be discussed in detail in Section 4.2.

---

**Algorithm 1** Dual-Path Condition Alignment Batch Step

---

1: **Input:** DDT $v_\theta$, batch of $B$ flow examples $F = \{(\mathbf{x}_1, y_1), \ldots, (\mathbf{x}_B, y_B)\}$, projector $z_\phi$, learning rate $\beta$, sampling times $K = 2$ and hyperparameter $\lambda = 0.5$.
2: **Output:** Updated model parameters $\theta$.
3: $L(\theta, \phi) = 0$
4: **for** $i$ in range($B$) **do**
5:    **for** $j$ in range($K$) **do**
6:       $t_j \sim U(0, 1)$, $\epsilon_j \sim \mathcal{N}(0, \mathbf{I})$, $\mathbf{x}_{t_j} = \alpha_{t_j} \mathbf{x}_i + \sigma_{t_j} \epsilon_j$
7:       $\hat{\mathbf{v}}_j, \mathbf{z}_j = v_\theta(\mathbf{x}_{t_j}, t_j, y_i)$, $\mathbf{v}_j = \dot{\alpha}_{t_j} \mathbf{x}_i + \dot{\sigma}_{t_j} \epsilon_j$
8:       $\mathbf{z}_j = z_\phi(\mathbf{z}_j)$
9:       $L(\theta, \phi) + = ||\hat{\mathbf{v}}_j - \mathbf{v}_j||^2$
10:       **for** $k$ in range($j$) **do**
11:          $L(\theta, \phi) - = \frac{2\lambda}{K(K-1)} \cdot \text{sim}(\mathbf{z}_k, \mathbf{z}_j)$
12:       **end for**
13:    **end for**
14: **end for**
15: $\theta \leftarrow \theta - \frac{\beta}{B} \nabla_\theta L(\theta, \phi)$, $\phi \leftarrow \phi - \frac{\beta}{B} \nabla_\phi L(\theta, \phi)$

---

## 4.2 CONDITION ALIGNMENT

In REPA and DDT, the features extracted from pure images by state-of-the-art visual encoders are used to align the conditional features learned by DiT blocks from noisy latents, which has been shown to significantly enhance the model's performance:

$$\mathcal{L}_{\text{REPA}}(\theta, \phi) = -\mathbb{E}_{\mathbf{x}_*, \epsilon, t} \left[ \frac{1}{N} \sum_{n=1}^{N} \text{sim}(\mathbf{y}_*^{[n]}, z_\phi(\mathbf{z}_t^{[n]})) \right] \tag{5}$$

where $\mathbf{y}_*$ denotes the output of the visual encoder, $\mathbf{z}_t$ represents the conditions extracted by DDT, and $z_\phi$ is a trainable MLP used to align the data dimensions of $\mathbf{y}_*$ and $\mathbf{z}_t$. $\theta$ and $\phi$ are the parameters of DDT and $z_\phi$, respectively. $N$ is the patch number and $\text{sim}(\cdot, \cdot)$ is a pre-defined similarity function.

However, large visual encoders introduce additional training data and model parameters. We posit that the features output by the visual encoder provide consistent and accurate conditioning for different noisy latents derived from the same pure image during training. The fact that different condition features of the same image converge toward the representation extracted by the visual encoder during training resembles *clustering* in unsupervised learning. This inspires us to sample multiple condition features in a single training step and align them towards the cluster center—which corresponds to the representation extracted by the visual encoder in REPA as intuitively illustrated in 2.

Similarly, We align any two conditions of $\{\mathbf{z}_{t_k}\}$ in the manner of REPA:

$$\mathcal{L}_{\text{DUPA}}(\theta, \phi) := -\mathbb{E}_{\mathbf{x}_*, \{\epsilon_k, t_k\}_{k=1}^{K}} \left[ \frac{2}{K(K-1)} \sum_{1 \le i < j \le K} \frac{1}{N} \sum_{n=1}^{N} \text{sim}(z_\phi(\mathbf{z}_{t_i}^{[n]}), z_\phi(\mathbf{z}_{t_j}^{[n]})) \right]. \tag{6}$$

On the other hand, we modify the original diffusion model's loss to the average of diffusion losses over $K$-times samplings:

$$\mathcal{L}_{\text{velocity}}(\theta) := \mathbb{E}_{\mathbf{x}_*, \{\epsilon_k, t_k\}_{k=1}^{K}} \left[ \sum_{k=1}^{K} \| \mathbf{v}_\theta(\mathbf{x}_{t_k}, t_k) - \dot{\alpha}_{t_k} \mathbf{x}_* - \dot{\sigma}_{t_k} \epsilon_k \|^2 \right]. \tag{7}$$

Then we sum the condition alignment loss and diffusion loss to construct the loss function for model training:

$$\mathcal{L} := \mathcal{L}_{\text{velocity}} + \lambda \mathcal{L}_{\text{DUPA}}, \tag{8}$$

where $\lambda$ is a hyperparameter that controls the tradeoff between condition alignment and denoising. Algorithm 1 illustrates the implementation of an arbitrary batch step in training DUPA.

## 5 EXPERIMENTS

We conduct extensive experiments to evaluate DUPA's performance and effectiveness, focusing on three key aspects:

- Performance comparison between DUPA and current state-of-the-art methods. (Section 5.2)

- Effectiveness and necessity of DUPA's components and settings. (Section 5.3, 5.4)

- Time and computational costs of DUPA during training and inference. (Section 5.5)

## 5.1 EXPERIMENTAL SETUP

**Implementation details.** Our experimental setup aligns with DiT, SiT, REPA, and DDT. DUPA is trained on $256 \times 256$ ImageNet datasets with a batch size of 256. Images are processed through the off-shelf Stable Diffusion VAE to obtain latents $\mathbf{z} \in \mathbb{R}^{32 \times 32 \times 4}$. Adam optimizer with a learning rate of 0.0001 is employed throughout the entire training process. DUPA's model configuration is shown in Appendix B, which maintains the same model size with SiT. We set hyperparameter $\lambda = 0.5$ and independent noise sampling times $K = 2$, choose cosine similarity as $\text{sim}(\cdot, \cdot)$ and do not use classifier-free guidance (CFG) unless otherwise specified. Our default training infrastructure consisted of 8×A100 GPUs. For more experimental details, please refer to Appendix D.

**Initialization of projector.** It is crucial to avoid setting both the weights and biases to 0 when initializing projector $z_\phi$. Otherwise, the condition used to align with will remain 0, leading to shortcut learning. In our experiments, we employ Kaiming initialization (He et al., 2015) for the first layer of projector $z_\phi$ to preserve variance during forward propagation, while utilizing a reduced-gain Xavier initialization (Glorot & Bengio, 2010) for subsequent layers to prevent gradient explosion or overfitting.

**Evaluation.** We report following five quantitative metrics to evaluate model's performance: Fréchet inception distance (FID; (Heusel et al., 2017)), sFID (Nash et al., 2021), inception score (IS; (Salimans et al., 2016)), precision (Prec.) and recall (Rec.) (Kynkäänniemi et al., 2019). We sample 50,000 images to calculate the above quantitative metrics.

**Sampler.** We use the SDE Euler-Maruyama sampler (for SDE with $w_t = \sigma_t$) and set the number of function evaluations (NFE) as 250 which follows SiT unless otherwise specified.

**Baselines.** We select state-of-the-art generative models in recent years as our baselines. Unlike other works, we do not distinguish DUPA and baselines based on model architecture, but rather based on the types of auxiliary tasks used for generation: (a) *No auxiliary task*: Dit (Peebles & Xie, 2023), SiT (Ma et al., 2024), FasterDiT (Yao et al., 2024) and DDT (Wang et al., 2025). (b) *Masked Image Modeling*: MaskGIT, (Chang et al., 2022), LlamaGen (Sun et al., 2024), VAR (Tian et al., 2024), MagViT-v2 (Yu et al., 2023), MAR (Li et al., 2024), MaskDiT (Zheng et al., 2024), MDT (Gao et al., 2023a) and MDTv2(Gao et al., 2023b). (c) *Contrastive learning*: ΔFM (Stoica et al., 2025) and Disp-Loss (Wang & He, 2025). (d) *Supervised representation alignment*: REPA (Yu et al., 2025). (e) *Unsupervised representation alignment*: DUPA. We categorize all autoregressive models as (b). The original DDT introduces architectural improvements, such as SwiGLU (Touvron et al., 2023), RoPE (Su et al., 2024), and RMSNorm (Touvron et al., 2023), as well as supervision from external visual encoders. Our approach solely focuses on its core contribution—decoupled encoder-decoder architecture. Therefore, the following results regarding DDT are all reproduced based on SiT.

## 5.2 SYSTEM-LEVEL COMPARISON

Table 3 shows the performance of our method compared to different sizes of base models. It can be seen that DUPA has improved all sizes of base models in various generation metrics.

Table 1 presents a comparative analysis of DUPA-XL/2 against current state-of-the-art methods on the ImageNet $256 \times 256$. In terms of sFID, DUPA outperforms all other listed methods, both with and without CFG. Furthermore, it achieves the best recall score in the non-CFG setting and the best precision score when CFG is applied.

Notably, for FID, DUPA surpasses all methods that do not rely on external supervision after only 400 training epochs. Even when compared to REPA—a model trained for a full 800 epochs with the aid of large visual encoders' representation alignment—DUPA's performance is within a narrow 3% margin. This achievement, despite the shorter training schedule (we train DUPA-XL/2 only for 400 epochs due to resource and time limits), strongly demonstrates the superior efficiency of DUPA.

Table 1: **System-Level Performance on ImageNet 256 × 256.** Our results are **bolded** to indicate that DUPA performs better than methods without external supervision of large visual encoders, while **highlighted** to indicate that DUPA performs the best among all methods. ↓ indicates a lower value is better and ↑ indicates a higher value is better.

| Method | Training Epochs | #params | External Images | External Params | Generation w/o CFG | | | | | Generation w/ CFG | | | | |
|---|---|---|---|---|---|---|---|---|---|---|---|---|---|---|
| | | | | | FID↓ | sFID↓ | IS↑ | Prec.↑ | Rec.↑ | FID↓ | sFID↓ | IS↑ | Prec.↑ | Rec.↑ |
| **No Auxiliary Task** | | | | | | | | | | | | | | |
| DiT | 1400 | 675M | 0 | 0 | 9.62 | 6.85 | 121.5 | 0.67 | 0.67 | 2.27 | 4.60 | 278.2 | 0.83 | 0.57 |
| SiT | 1400 | 675M | 0 | 0 | 8.61 | 6.32 | 131.7 | 0.68 | 0.67 | 2.06 | 4.50 | 270.3 | 0.82 | 0.59 |
| FasterDiT | 400 | 675M | 0 | 0 | 7.91 | 5.45 | 131.3 | 0.67 | **0.69** | 2.03 | 4.63 | 264.0 | 0.81 | 0.60 |
| DDT | 400 | 675M | 0 | 0 | 8.06 | 5.31 | 127.4 | 0.69 | 0.67 | 2.01 | 4.66 | 281.7 | 0.80 | 0.59 |
| **Masked Image Modeling** | | | | | | | | | | | | | | |
| MaskGIT | 555 | 227M | 0 | 0 | 6.18 | - | 182.1 | **0.80** | 0.51 | - | - | - | - | 0.58 |
| LlamaGen | 300 | 3.1B | 0 | 0 | 9.38 | 8.24 | 112.9 | 0.69 | 0.67 | 2.18 | 5.97 | 263.3 | 0.81 | 0.58 |
| VAR | 350 | 2.0B | 0 | 0 | - | - | - | - | - | 1.80 | - | 365.4 | 0.83 | 0.57 |
| MagViT-v2 | 1080 | 307M | 0 | 0 | 3.65 | - | 200.5 | - | - | 1.78 | - | 319.4 | - | - |
| MAR | 800 | 945M | 0 | 0 | **2.35** | - | **227.8** | 0.79 | 0.62 | 1.55 | - | 303.7 | 0.81 | 0.62 |
| MaskDiT | 1600 | 675M | 0 | 0 | 5.69 | 10.34 | 177.9 | 0.74 | 0.60 | 2.28 | 5.67 | 276.6 | 0.80 | 0.61 |
| MDT | 1300 | 675M | 0 | 0 | 6.23 | 5.23 | 143.0 | 0.71 | 0.65 | 1.79 | 4.57 | 283.0 | 0.81 | 0.61 |
| MDTv2 | 920 | 675M | 0 | 0 | - | - | - | - | - | 1.58 | 4.52 | 314.7 | 0.79 | 0.65 |
| **Contrastive Learning** | | | | | | | | | | | | | | |
| ΔFM | 800 | 675M | 0 | 0 | - | - | - | - | - | 1.97 | 4.53 | 268.4 | 0.79 | 0.65 |
| Disp-Loss | 1200 | 675M | 0 | 0 | - | - | - | - | - | 1.97 | 4.61 | 275.2 | 0.80 | 0.63 |
| **Supervised Representation Alignment** | | | | | | | | | | | | | | |
| | 80 | | | | 7.90 | 5.06 | 122.6 | 0.70 | 0.65 | - | - | - | - | - |
| REPA | 200 | 675M | 142M | 1.1B | 6.40 | - | - | - | - | 1.96 | 4.49 | 264.0 | 0.82 | 0.60 |
| | 800 | | | | 5.90 | 5.73 | 157.8 | 0.70 | **0.69** | 1.42 | 4.70 | 305.7 | 0.80 | 0.65 |
| **Unsupervised Representation Alignment** | | | | | | | | | | | | | | |
| | 80 | | | | 8.71 | 4.65 | 114.6 | 0.70 | 0.65 | 2.28 | 4.48 | 237.2 | 0.83 | 0.59 |
| **DUPA (Ours)** | 200 | 675M | 0 | 0 | 6.57 | 4.63 | 136.5 | 0.70 | 0.68 | 1.70 | 4.45 | 265.3 | 0.83 | 0.61 |
| | 400 | | | | 5.92 | **4.63** | 149.6 | 0.71 | **0.69** | **1.46** | **4.45** | 296.2 | **0.84** | 0.62 |

Table 2: **Component-wise analysis.** All models are DUPA-L/2 trained for 400K iterations with different settings. "Resampling" column indicates whether to independently resample timestamp $t$ or noise $\epsilon$.

| Resampling | Depth | Objective | $\lambda$ | FID↓ |
|---|---|---|---|---|
| Vanilla SiT-L/2 | | | | 18.8 |
| $t$ | 8 | Cos. sim. | 0.5 | 13.2 |
| $\epsilon$ | 8 | Cos. sim. | 0.5 | 12.4 |
| $t, \epsilon$ | 4 | Cos. sim. | 0.5 | 11.8 |
| $t, \epsilon$ | 6 | Cos. sim. | 0.5 | 11.3 |
| $t, \epsilon$ | 10 | Cos. sim. | 0.5 | 11.2 |
| $t, \epsilon$ | 12 | Cos. sim. | 0.5 | 11.6 |
| $t, \epsilon$ | 14 | Cos. sim. | 0.5 | 11.9 |
| $t, \epsilon$ | 16 | Cos. sim. | 0.5 | 12.1 |
| $t, \epsilon$ | 8 | NT-Xent | 0.5 | 11.6 |
| $t, \epsilon$ | 8 | Cos. sim. | 0.25 | 11.2 |
| $t, \epsilon$ | 8 | Cos. sim. | 0.75 | 11.1 |
| $t, \epsilon$ | 8 | Cos. sim. | 1 | 11.1 |
| $t, \epsilon$ | 8 | Cos. sim. | 0.5 | 11.1 |

Table 3: Model performance across different sizes with 400K training steps.

| Model | FID↓ | sFID↓ | IS↑ | Prec.↑ | Rec.↑ |
|---|---|---|---|---|---|
| SiT-B/2 | 33.0 | 6.46 | 43.7 | 0.53 | 0.63 |
| DDT-B/2 | 29.5 | 6.23 | 51.7 | 0.57 | 0.63 |
| **DUPA-B/2** | **25.2** | **5.89** | **67.4** | **0.61** | **0.63** |
| SiT-L/2 | 18.8 | 5.29 | 72.0 | 0.64 | 0.64 |
| DDT-L/2 | 14.9 | 5.17 | 87.8 | 0.65 | 0.64 |
| **DUPA-L/2** | **11.1** | **4.91** | **104.8** | **0.69** | **0.65** |
| SiT-XL/2 | 17.2 | 5.07 | 76.5 | 0.65 | 0.63 |
| DDT-XL/2 | 12.8 | 4.98 | 91.3 | 0.67 | 0.63 |
| **DUPA-XL/2** | **8.71** | **4.65** | **114.6** | **0.70** | **0.65** |

Table 4: Ablation study of proposed improvements.

| Method | FID↓ | sFID↓ | IS↑ | Prec.↑ | Rec.↑ |
|---|---|---|---|---|---|
| DDT-L/2 | 14.9 | 5.17 | 87.8 | 0.65 | 0.64 |
| + *Dual-Path Sampling* | 12.5 | 5.02 | 96.6 | 0.68 | 0.65 |
| + *Condition Alignment* | 11.1 | 4.91 | 104.8 | 0.69 | 0.65 |

## 5.3 COMPONENT-WISE ANALYSIS

The resampling strategy, encoder-decoder architecture, condition alignment method and hyperparameter settings of DUPA significantly impact the model's performance. Results of the impact of these components are shown in Table 2.

**Resampling strategy.** Experiments show that independently resampling of both timestamp $t$ and noise $\epsilon$ performs the best. We believe this provides more diverse noisy images, thereby enhancing the reliability of cluster centers of extracted condition representations.

**Condition encoder depth.** We investigate the impact of the number of layers in the condition encoder on DUPA-L/2. Similar to the conclusion in REPA, aligning the representations output by

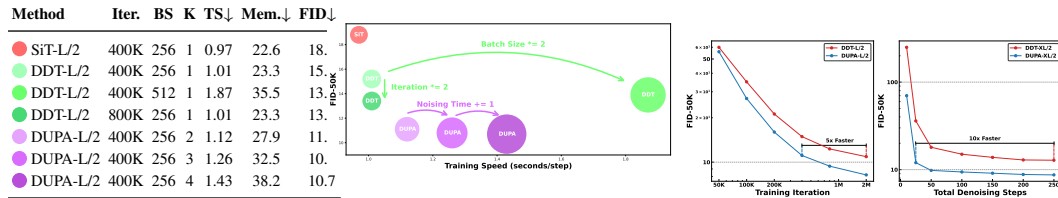

| Method | Iter. | BS | K | TS↓ | Mem.↓ | FID↓ |
|--------|-------|-----|---|-----|-------|------|
| ● SiT-L/2 | 400K | 256 | 1 | 0.97 | 22.6 | 18. |
| ● DDT-L/2 | 400K | 256 | 1 | 1.01 | 23.3 | 15. |
| ● DDT-L/2 | 400K | 512 | 1 | 1.87 | 35.5 | 13. |
| ● DDT-L/2 | 800K | 256 | 1 | 1.01 | 23.3 | 13. |
| ● DUPA-L/2 | 400K | 256 | 2 | 1.12 | 27.9 | 11. |
| ● DUPA-L/2 | 400K | 256 | 3 | 1.26 | 32.5 | 10. |
| ● DUPA-L/2 | 400K | 256 | 4 | 1.43 | 38.2 | 10.7 |

(a) "BS" indicates batch size, "K" indicates noising times, "TS" indicates training speed (sec/step) and "Mem." indicates memory usage of a single GPU (GB).

(b) Image sampling is performed on DUPA-XL/2 and DDT-XL/2 trained for 400K iterations.

Figure 3: **Time and computational cost analysis.** (a) Time and computational costs comparison. (b)Training efficiency and inference speed comparison.

the first few layers can help the subsequent network predict high-frequency details. In the remaining experiments, we perform condition alignment at the $8th$ layer.

**Alignment objective.** We compare the effects of two similarity functions which are commonly used in contrastive learning: Normalized Temperature-scaled Cross Entropy (NT-Xent) and negative cosine similarity (cos. sim.), and we choose cos. sim. in other experiments.

**Effect of tradeoff parameter.** As shown in Table 2, DUPA is robust to the tradeoff parameter $\lambda$.

## 5.4 ABLATION STUDY

Compared to the baseline model DDT, our primary improvements lie in dual-path sampling and condition alignment. Since condition alignment relies on dual-path sampling, we conduct the following three sets of ablation experiments on DUPA-L/2: DUPA without dual-path sampling and condition alignment (which degenerates to DDT), DUPA without condition alignment and the vanilla DUPA. The results of the ablation experiments are shown in Table 4.

Dual-path sampling offers more precise gradient guidance for model parameter optimization in a training step, enhancing training efficiency, while conditional alignment enables the condition encoder to capture more accurate semantic representations from noisy images, further boosting model performance.

## 5.5 TIME AND COMPUTATIONAL COSTS

Since training and sampling of generative models require significant time and computational resources, we emphasize evaluating the model's computational cost in addition to its performance. During training, multiple independent sampling of noises and velocity prediction for single image represent the primary extra computational overhead introduced by our method. For the sampling phase, we also conduct experiments to explore whether DUPA can accelerate the sampling procedure through aligned condition feature.

**Noise sampling times.** We compare the impact of different noise sampling times on training speed, GPU memory usage, and model performance in Figure 3a. To illustrate the difference between multiple sampling and batch size enlargement, we additionally train DDT with a batch size of $2 \times 256 = 512$.

Neither doubling the batch size nor the training steps of DDT can achieve the performance of DUPA. Moreover, the former approach leads to a nearly doubled training cost. On the other hand, increasing $K$ significantly raises GPU memory usage and slows down training speed, without significant FID gains. We thus select $K = 2$ in other experiments.

**Improved training efficiency and inference speed.** To accelerate experiments, we compare the training efficiency of DUPA-L/2 and DDT-L/2. As shown in Figure 3b, DUPA achieved $5\times$ and $10\times$ speedups over DDT in training and inference stages respectively without external supervision, which highlights DUPA's remarkable efficiency.

## 6 CONCLUSION AND FUTURE WORK

Inspired by REPA, we propose DUPA, which provides efficient semantic information for denoising-based generative models' training by aligning the representations of different noisy views from the same image, which is similar to REPA. DUPA can achieve performance comparable to that of REPA without any external supervision of large visual encoder, which can easily applied to any denoising-based models. Furthermore, we intend to conduct further testing and improvement of DUPA on text-to-image tasks in the future.

ACKNOWLEDGMENT

This work was supported by National Science and Technology Major Project (2024ZD01NL00101), Natural Science Foundation of China (62271013), Guangdong Provincial Key Laboratory of Ultra High Definition Immersive Media Technology (2024B1212010006), Guangdong Province Pearl River Talent Program (2021QN020708), Guangdong Basic and Applied Basic Research Foundation (2024A1515010155), Shenzhen Science and Technology Program (JCYJ20240813160202004, JCYJ20230807120808017, SYSPG20241211173440004), and was also financially supported for Outstanding Talents Training Fund in Shenzhen.

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

## A    USE OF LARGE LANGUAGE MODELS

We acknowledge the use of Large Language Models (LLMs), specifically OpenAI's GPT-5 and Google's Gemini 2.5 Pro, to assist in the preparation of this manuscript. The specific applications were as follows:

- **Information Gathering:** To assist in consulting background information and identifying potential literature related to the research field.
- **Language and Readability:** To improve the grammar, clarity, and overall readability of the manuscript through language polishing.
- **Format Checking:** To assist in checking the paper's layout and citation style for general compliance with conference requirements.

We emphasize that all scientific claims, cited works, experimental results, and final conclusions were independently reviewed and verified by the human authors. The authors take full and final responsibility for the entire content of this submission, including any potential errors or inaccuracies, in accordance with ICLR policy.

## B    MODEL CONFIGURATION

Table 5: Model configuration details.

| Config | #Layers | Hidden dim | #Heads | Enc depth | Patch size |
|---|---|---|---|---|---|
| DUPA-S/2 | 12 | 384 | 6 | 4 | 2 |
| DUPA-B/2 | 12 | 768 | 12 | 4 | 2 |
| DUPA-L/2 | 24 | 1024 | 16 | 8 | 2 |
| DUPA-XL/2 | 28 | 1152 | 16 | 8 | 2 |

## C    CLASSIFIER FREE GUIDANCE

Considering that classifier-free guidance can significantly affect the generation quality, we adopt interval guidance with interval $[0, 0.7]$ following REPA, which apply classifier-free guidance only to the phase of generating high-frequency details, thereby ensuring the diversity of the generation results. The results of classifier-free guidance scale $w$ are shown in Table 6.

Table 6: Detailed evaluation results of DUPA-XL/2 at 2M iteration with different classifier-free guidance scale $w$.

| Model | #Params | Iter. | $w$ | FID↓ | sFID↓ | IS↑ | Prec.↑ | Rec.↑ |
|---|---|---|---|---|---|---|---|---|
| DUPA-XL/2 | 675M | 2M | 1.56 | 1.51 | 4.47 | 274.6 | 0.82 | 0.63 |
| DUPA-XL/2 | 675M | 2M | 1.58 | 1.47 | 4.45 | 286.8 | 0.83 | 0.62 |
| DUPA-XL/2 | 675M | 2M | 1.60 | 1.46 | 4.45 | 296.2 | 0.84 | 0.62 |
| DUPA-XL/2 | 675M | 2M | 1.62 | 1.49 | 4.44 | 304.7 | 0.84 | 0.61 |
| DUPA-XL/2 | 675M | 2M | 1.64 | 1.53 | 4.43 | 309.5 | 0.84 | 0.60 |

# D  IMPLEMENTATION DETAILS

Table 7: Experimental setup.

| | Table 1 (DUPA-XL/2) | Table 2 (DUPA-L/2) | Table 4 (DUPA-L/2) | Figure 3a (DUPA-L/2) |
|---|---|---|---|---|
| **Architecture** | | | | |
| Input dim. | $32{\times}32{\times}4$ | $32{\times}32{\times}4$ | $32{\times}32{\times}4$ | $32{\times}32{\times}4$ |
| Num. layers | 28 | 24 | 24 | 24 |
| Hidden dim. | 1,152 | 1,024 | 1,024 | 1,024 |
| Num. heads | 16 | 16 | 16 | 16 |
| **DUPA** | | | | |
| $\lambda$ | 0.5 | 0.25$\sim$1 | 0.5 | 0.5 |
| Alignment depth | 8 | 4$\sim$16 | 8 | 8 |
| sim$(\cdot, \cdot)$ | cos. sim. | cos. sim./NT-Xent | cos. sim. | cos. sim. |
| Noising Times | 2 | 2 | 2 | 2$\sim$4 |
| **Optimization** | | | | |
| Training iteration | 2M | 400K | 400K | 400K |
| Batch size | 256 | 256 | 256 | 256 |
| Optimizer | AdamW | AdamW | AdamW | AdamW |
| lr | 0.0001 | 0.0001 | 0.0001 | 0.0001 |
| $(\beta_1, \beta_2)$ | (0.9, 0.999) | (0.9, 0.999) | (0.9, 0.999) | (0.9, 0.999) |
| **Interpolants** | | | | |
| $\alpha_t$ | $1-t$ | $1-t$ | $1-t$ | $1-t$ |
| $\sigma_t$ | $t$ | $t$ | $t$ | $t$ |
| $w_t$ | $\sigma_t$ | | $\sigma_t$ | $\sigma_t$ |
| Training objective | v-prediction | v-prediction | v-prediction | v-prediction |
| Sampler | Euler-Maruyama | Euler-Maruyama | Euler-Maruyama | Euler-Maruyama |
| Sampling steps | 250 | 250 | 250 | 250 |
| Guidance | 1.6 | - | - | - |

# E  DISCRIMINATIVE SEMANTICS

Figure 4 presents a comprehensive discriminative semantics analysis of the DUPA-XL/2 and SiT-XL/2 models, evaluated through two key metrics: linear probing validation accuracy and CKNNA score.

**Linear probing.** The linear probing results in Figure 4a show that both DUPA-XL/2 and SiT-XL/2 models exhibit an initial increase in validation accuracy as layer depth increases, before eventually plateauing or decreasing. This trend is typical for discriminative models, where the initial layers learn basic features and the later layers learn more abstract, task-specific features.

Significantly, the DUPA-XL/2 model consistently outperforms SiT-XL/2 across all layers. At its peak performance, DUPA-XL/2 achieves 69% validation accuracy, while the SiT-XL/2 model peaks at 53.5%. This large performance gap highlights DUPA-XL/2's superior ability to learn more discriminative, semantically rich representations.

**CKNNA score.** As shown in 4b, DUPA-XL/2 demonstrates a much higher CKNNA score than the SiT-XL/2 across all three time steps ($t$=0.0, $t$=0.25, and $t$=0.5). CKNNA score, which measures the complexity and discriminative power of the learned features, is consistently over 0.4 for DUPA-XL/2, whereas SiT-XL/2's score remains below 0.2.

This result indicates that the features extracted by DUPA-XL/2 are not only more discriminative but also more complex and better structured for classification tasks compared to those of SiT-XL/2. The consistent gap in CKNNA scores across different time steps further confirms that the superior discriminative capability of DUPA-XL/2 is a robust characteristic of the model's architecture.

# F  ALIGNMENT LOSS

Figure 5 shows the change in cosine similarity during DUPA-XL/2 training, measured across different denoising paths for condition alignment. Initially, most of the network's neurons are not activated, which leads to similar yet uninformative representations (note the initialization of the projector $z_\phi$ to prevent shortcut learning). In the early stage of training, DUPA begins to learn image features, but the cosine similarity rapidly decreases due to the influence of noise. After a small

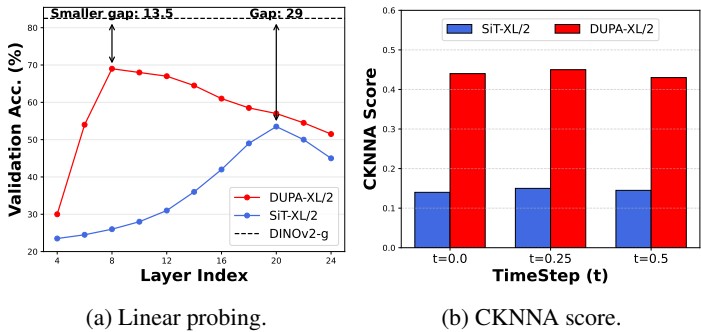

(a) Linear probing.

(b) CKNNA score.

Figure 4: **Discriminative semantics analysis.**

number of training steps (approximately 3,000 steps), DUPA begins to learn useful representations from different noisy latents of the same image, *i.e.*, the invariant semantic information from the pure image. Subsequently, the cosine similarity increases as training progresses.

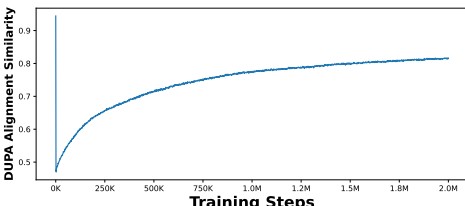

Figure 5: DUPA alignment similarity during training.

# G    MORE QUALITATIVE RESULTS

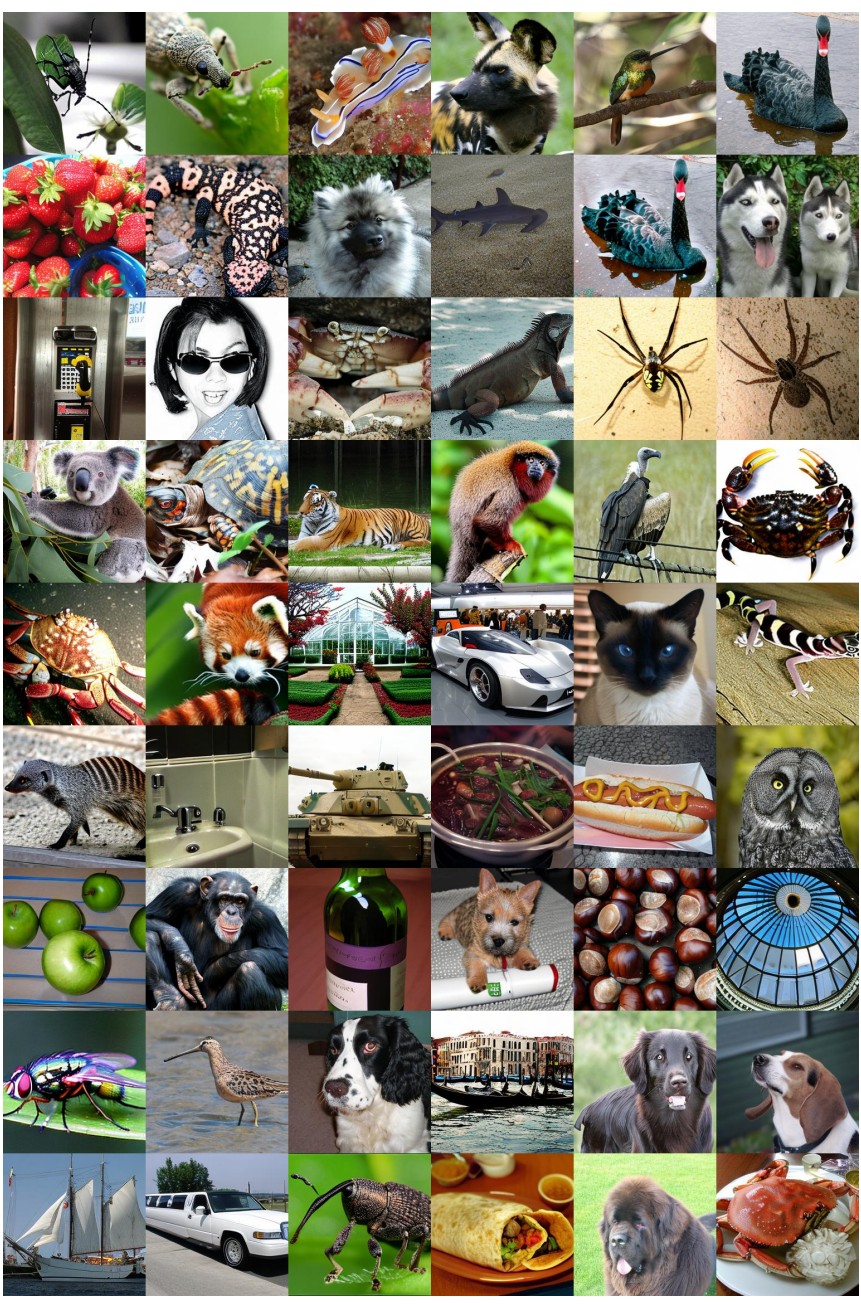

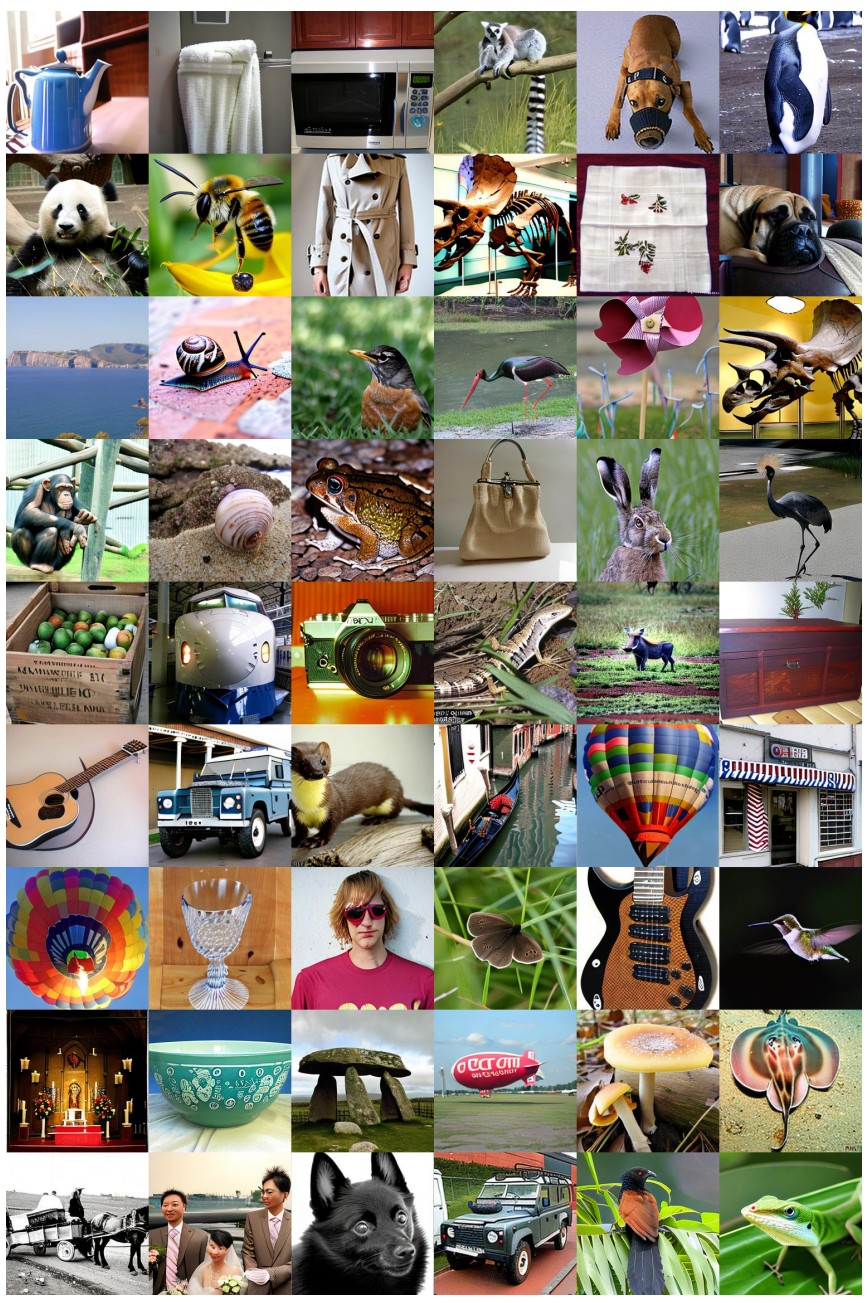

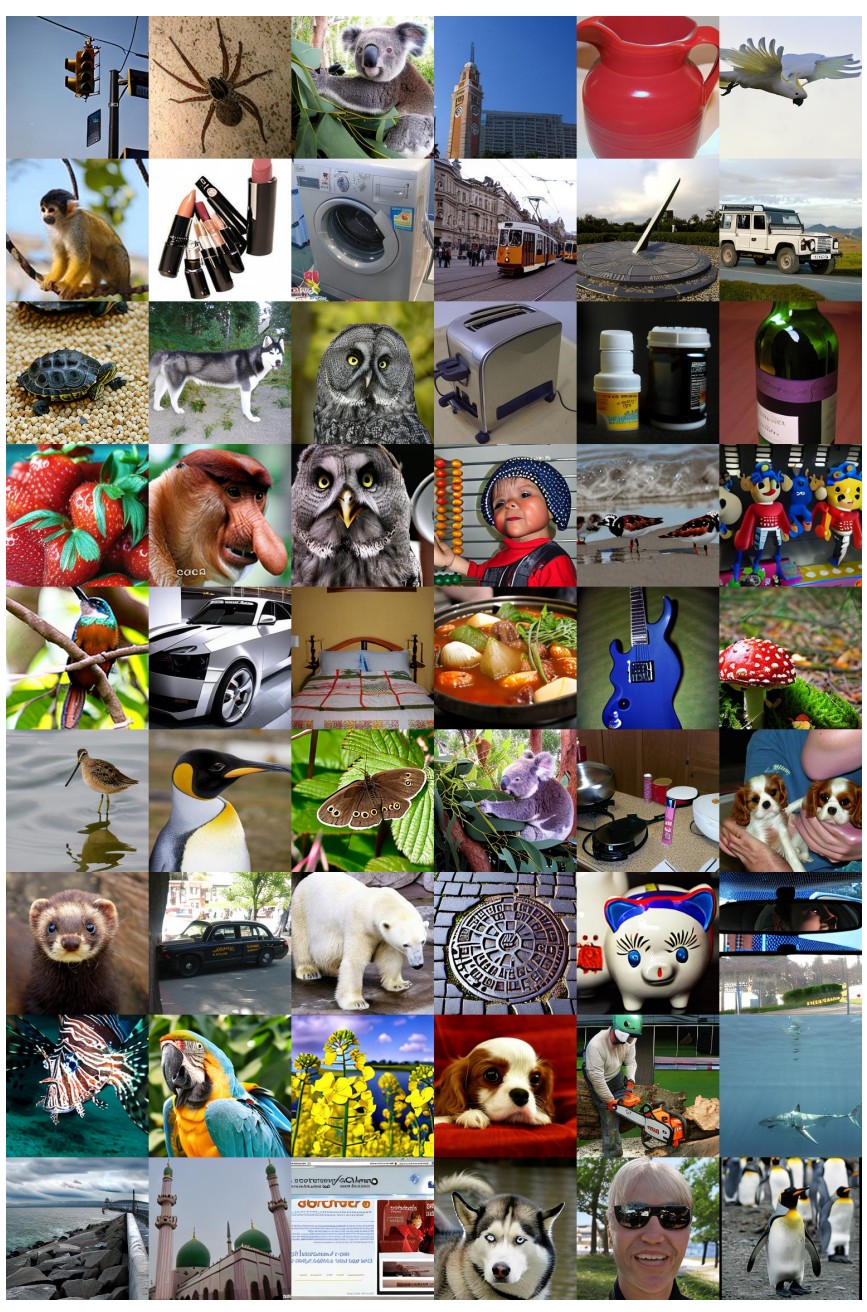

