# OpenReview forum: "Dual-Path Condition Alignment for Diffusion Transformers"
_ICLR.cc/2026/Conference — ICLR 2026 Poster_

### Official Review · Reviewer_A859 · 2025-10-31

**Soundness:** 3
**Presentation:** 3
**Contribution:** 3
**Rating:** 6
**Confidence:** 4

**Summary:**

This paper aims to accelerate the diffusion transformer training without an external pre-trained encoder by aligning latents from multiple noisy images through decoupled diffusion transformers. They show the proposed method, DUPA, can perform better than REPA without any external guidance.

**Strengths:**

1. The paper is well-motivated and easy-to-follow.

2. The proposed method, DUPA, consistently improves the baselines, and even perform better than REPA.

3. Extensive analysis demonstrates the effectiveness of each suggested component.

**Weaknesses:**

1. The authors point out the weakness of REPA in practical scenarios (i.e., beyond the ImageNet), e.g., out-of-distribution problem can arise, and additional pretraining is required. However, the authors do not handle such scenarios in the main experiments: They only conducted ImageNet generation results, which makes it difficult to argue whether the proposed method indeed addresses the problem of REPA. I think such a problem of REPA does not arise in the image generation problem, as (1) recent pretrained visual representations (e.g., DINOv3 and SigLIPv2) are trained on extremely large-scale datasets, and (2) we can easily use open-source image encoders.

2. The authors fixed the batch size at 256 for their experiments, but I think that increasing the sampling times K can have a similar effect to enlarging the batch size. In fact, a previous study [1] has shown that applying augmentations to the same batch can yield better performance with fewer iterations. Therefore, it is needed to verify how much of the improvement in the proposed method comes from this effect, e.g., by training SiT with a batch size of 512, or using sampling times K with only a flow matching loss.

[1] Hoffer et al., Augment Your Batch: Improving Generalization Through Instance Repetition, CVPR 2020

**Questions:**

Please answer the Weaknesses.

---

> ### Author Response · Authors · 2025-11-21
> **Response to Reviewer A859**
>
> We sincerely appreciate your valuable time and effort spent reviewing our paper. Our response to your question is as follows:
>
> $\textbf{[Q1]}$ The authors point out the weakness of REPA in practical scenarios (i.e., beyond the ImageNet), e.g., out-of-distribution problem can arise, and additional pretraining is required. However, the authors do not handle such scenarios in the main experiments: They only conducted ImageNet generation results, which makes it difficult to argue whether the proposed method indeed addresses the problem of REPA. I think such a problem of REPA does not arise in the image generation problem, as (1) recent pretrained visual representations (e.g., DINOv3 and SigLIPv2) are trained on extremely large-scale datasets, and (2) we can easily use open-source image encoders.
>
> $\textbf{[A1]}$ In our opinion, when applying generative models such as SiT and DDT to vertical-domain generation tasks (e.g., generating depth maps for specific scenes), external visual encoders may not necessarily provide effective representation guidance. From our perspective, in such highly specialized tasks, the representation spaces provided by external general-purpose visual encoders (e.g., DINOv3, SigLIPv2) may not align well with the effective representations required for the specific domain, potentially leading to suboptimal guidance. In such cases, our proposed self-alignment approach can learn more task-relevant representations directly from the domain data, thereby underscoring its unique advantages.
>
> We fully agree with your suggestion that additional relevant experiments would strengthen our argument. We are actively preparing such experiments and will provide timely updates if progress is made.
>
> $\textbf{[Q2]}$ The authors fixed the batch size at 256 for their experiments, but I think that increasing the sampling times K can have a similar effect to enlarging the batch size. In fact, a previous study has shown that applying augmentations to the same batch can yield better performance with fewer iterations. Therefore, it is needed to verify how much of the improvement in the proposed method comes from this effect, e.g., by training SiT with a batch size of 512, or using sampling times K with only a flow matching loss.
>
> $\textbf{[A2]}$ Figure 3a in our paper demonstrates the impact of increasing DDT's training batch size to 512 or doubling the training iterations. We present the results again here for clarity. Please note that $\textbf{DUPA and DDT share the same model architecture. When dual-path sampling is not used (sampling time=1, which also precludes condition alignment), DUPA degenerates into DDT}$. Results indicate that dual-path sampling is more efficient, which we attribute to applying multiple noisings to a single image, enabling the model to extract information more effectively from clean images.
>
> | Method        | Iterations | Batch Size | Sampling Times | FID↓ |
> | ------------- | ---------- | ---------- | -------------- | ---- |
> | SiT-L/2       | 400K       | 256        | 1              | 18.8 |
> | DDT(DUPA)-L/2 | 400K       | 256        | 1              | 15.2 |
> | DDT(DUPA)-L/2 | 400K       | 512        | 1              | 13.9 |
> | DDT(DUPA)-L/2 | 800K       | 256        | 1              | 13.4 |
> | DUPA-L/2      | 400K       | 256        | 2              | 11.1 |
> | DUPA-L/2      | 400K       | 256        | 3              | 10.8 |
> | DUPA-L/2      | 400K       | 256        | 4              | 10.7 |

---

> ### Author Response · Authors · 2025-11-23
> **Experiments on ChestX-ray8**
>
> ChestX-ray8 dataset[1] contains more than 100,000 medical images from tens of thousands of patients, annotated with common thoracic diseases such as atelectasis, cardiomegaly, effusion, etc. We conduct 256×256-resolution image-generation experiments on ChestX-ray 8. DUPA-L/2, REPA-L/2 and SiT-L/2 are each trained for 80 epochs. We sample 50,000 images without CFG. The results are as follows (without CFG):
>
> | Model    | FID↓  | sFID↓ | Pre.↑ | Rec.↑|
> | -------- | ---- | ---- | ---- | ---- |
> | SiT-L/2  | 23.7 | 31.3 | 0.77 | 0.74 |
> | REPA-L/2 | 19.4 | 26.9 | 0.81 | 0.76 |
> | DUPA-L/2 | 16.1 | 24.2 | 0.82 | 0.76 |
>
>
> We omit Inception Score because its core assumption—“a generator should produce sharp and diverse images so that Inception-v3 can assign each sample to a single class with high confidence while the overall distribution remains uniform”—does not hold for medical images. Chest X-rays, CT and MRI scans are grey-scale, low-texture, and visually similar across classes; Inception-v3 can hardly distinguish them (the resulting IS is only 2\~3 and is not informative).
>
> As shown in the table, REPA still improves performance on images with specific distributions in the vertical domain, but its gain is now smaller than that of DUPA, which shows the efficiency and transferability of DUPA.
>
> [1] Wang, Xiaosong, et al." ChestX-ray8: Hospital-scale Chest X-ray Database and Benchmarks on Weakly-Supervised Classification and Localization of Common Thorax Diseases." arXiv preprint arXiv:1705.02315 (2017).

---

> > ### Comment · Reviewer_A859 · 2025-11-27
> >
> > Thank you for the rebuttal. The replies have addressed my concerns. Thus, I will raise the rating to 8 and recommend acceptance.

---

### Official Review · Reviewer_7oUe · 2025-11-01

**Soundness:** 3
**Presentation:** 3
**Contribution:** 2
**Rating:** 4
**Confidence:** 5

**Summary:**

This work (DUPA) first points out the internal issues brought by REPA:
(1) Out of distribution. and (2) Huge additional computational costs.
Then inspired by the idea of DDT, SD-DIT and Contrastive FM works, the authors propose DUPA to directly align  two noisy views of a single image without external supervision. And such self-alignment (unsupervised alignment)  can significantly accelerate the convergence of SiT.

**Strengths:**

1. The proposed DUPA is simple but effective, without the reliance of external ViTs, like dinov2.
2. The author’s writing is very straightforward and concise, without storytelling or beating around the bush.
3. Clear convergence of SiT is brought by DUPA.
4. Sufficient experiments .

**Weaknesses:**

1. More discussion and analysis about: why such self-supervised alignment could work for the convergence of SiT?
for example in SD-DiT[2], the choice of t=min (mostly close to pure image) is the most effective acceleration technique. And how about DUPA?
2.  Recently there are some works [1][2] like DUPA, focusing on the self-alignment DiT, please claim the difference/advantage/difsussion compared with DUPA  and add the corresponding reference.

[1] Jiang, Dengyang, et al. "No Other Representation Component Is Needed: Diffusion Transformers Can Provide Representation Guidance by Themselves." arXiv preprint arXiv:2505.02831 (2025).
[2] Zhu, Rui, et al. "Sd-dit: Unleashing the power of self-supervised discrimination in diffusion transformer." Proceedings of the IEEE/CVF Conference on Computer Vision and Pattern Recognition. 2024.

**Questions:**

See weakness.

I am very willing to raise my ratings if you can provide sufficient discussions mentioned in Weaknesses.

---

> ### Author Response · Authors · 2025-11-21
> **Response to Reviewer 7oUe (1/2)**
>
> We sincerely appreciate your valuable time and effort spent reviewing our paper. Our response to your question is as follows:
>
> $\textbf{[Q1]}$ More discussion and analysis about: why such self-supervised alignment could work for the convergence of SiT? for example in SD-DiT, the choice of t=min (mostly close to pure image) is the most effective acceleration technique. And how about DUPA?
>
> $\textbf{[A1]}$ The performance gains of DUPA originate from two key aspects:
>
> $\textbf{Dual-Path Sampling}$: Multiple noisings to a single image enable the model to more effectively extract information from clean images. As demonstrated in Figure 3a in our paper, this approach proves more efficient than increasing batch size or training iterations.
>
> $\textbf{Condition Alignment}$: We can obtain semantic information (condition) regarding the same clean image across different paths and timesteps through DDT. Aligning these conditions encourages DDT to extract more accurate semantic conditions, thereby further enhancing model performance.
>
> To investigate the impact of timestep selection, we conduct experiments under three configurations: using only dual-path sampling, using only condition alignment, and using both improvements simultaneously. For experimental efficiency, we conduct tests on ImageNet at 256×256 resolution using DUPA-B/2, training for 80 epochs without using CFG. Time intervals in the table below denote the range from which t is sampled for one branch of dual-path sampling, while the other branch retains the original sampling strategy. We use uniform sampling across the time interval.
>
> 1. Only dual-path sampling.
>
> | Time Interval | FID-50K |
> | ------------- | ------- |
> | \[0,0.1)      | 28.92   |
> | \[0,0.2)      | 28.14   |
> | \[0,0.3)      | 27.45   |
> | \[0,1)        | 26.21   |
>
> A broader sampling range enables the model to encounter more diverse intermediate states $z_t$, thereby enhancing performance.
>
> 2. Only condition alignment.
>
> To investigate the impact of timestep selection on condition alignment, we apply the stop-gradient operation to one branch of the dual-path sampling (which can be regarded as the teacher branch), utilizing only the intermediate conditions output by the teacher branch for condition alignment without computing the diffusion loss of the teacher branch.
>
> | Time Interval | FID-50K |
> | ------------- | ------- |
> | \[0,0.1)      | 27.21   |
> | \[0,0.2)      | 27.04   |
> | \[0,0.3)      | 27.13   |
> | \[0,1)        | 28.17   |
> | \[0.8,1)      | 30.36   |
>
> Selecting a relatively small t (closer to the clean image) in the teacher branch is most beneficial for model performance. This is understandable because when the teacher branch is frozen, its output effectively serves as "ground truth" that guides the model. Inaccurate outputs generated from large t (blurred images) would harm model performance.
>
> 3. Both dual-path sampling and condition alignment.
>
> | Time Interval | FID-50K |
> | ------------- | ------- |
> | \[0,0.1)      | 26.17   |
> | \[0,0.2)      | 25.72   |
> | \[0,0.3)      | 25.58   |
> | \[0,1)        | 25.23   |
>
> When both improvements are adopted simultaneously, the result of not restricting the selection of t is better, which also reflects the simplicity of the proposed method as it does not require too much manual configuration.

---

> ### Author Response · Authors · 2025-11-21
> **Response to Reviewer 7oUe (2/2)**
>
> $\textbf{[Q2]}$ Recently there are some works (SRA, SD-DiT) like DUPA, focusing on the self-alignment DiT, please claim the difference/advantage/difsussion compared with DUPA and add the corresponding reference.
>
> $\textbf{[A2]}$ Compared to the works you have listed and to our knowledge, DUPA shows the following differences and advantages in our opinion:
>
> $\textbf{Dual-Path Sampling}$: It not only enhances base model performance but also provides intermediate representations for alignment.
>
> $\textbf{Using DDT as Base Model}$: DDT identifies that early DiT layers extract low-frequency semantic information and structurally decouples them [1]. This perspective and approach align precisely with our premise: given a clean image, the low-frequency semantic information used to predict the current velocity should be consistent with the original image across different paths (initial noise) and timesteps, and we can align conditions extract by DDT to improve model performance.
>
> $\textbf{Simplicity, Efficiency and Transferability}$: Compared to SD-DiT[2], MaskDiT[3] and MAGE[4], DUPA requires no specially designed mask strategy; compared to SRA[5] and SD-DiT[2], DUPA does not need a specially designed noising strategy for the teacher branch.As shown below, DUPA demonstrates better performance than the above methods. These advantages make DUPA easier to adapt to other tasks (e.g., text-to-image) and datasets.
>
> | Method  | Training Epochs | FID-50K |
> | ------- | --------------- | ------- |
> | DUPA    | 400             | 1.46    |
> | SD-DiT  | 480             | 3.23    |
> | SRA     | 800             | 1.58    |
> | MaskDiT | 1600            | 2.28    |
> | MAGE    | 1600            | 7.04    |
>
> [1] Wang, Shuai, et al. "DDT: Decoupled Diffusion Transformer." arXiv preprint arXiv:2504.05741 (2025).
>
> [2] Zhu, Rui, et al. "SD-DiT: Unleashing the power of self-supervised discrimination in diffusion transformer." Proceedings of the IEEE/CVF Conference on Computer Vision and Pattern Recognition. 2024.
>
> [3] Zheng, Hongkai, et al. "Fast Training of Diffusion Models with Masked Transformers." arXiv preprint arXiv:2306.09305 (2023).
>
> [4] Li, Tianhong, et al. "MAGE: MAsked Generative Encoder to Unify Representation Learning and Image Synthesis" 2211.09117 (2022).
>
> [5] Jiang, Dengyang, et al. "No Other Representation Component Is Needed: Diffusion Transformers Can Provide Representation Guidance by Themselves." arXiv preprint arXiv:2505.02831 (2025).

---

> > ### Comment · Reviewer_7oUe · 2025-11-26
> >
> > Thanks for your replyment and sufficient exps from issues, which solved my questions.
> > Please these addtional exps/analysis in your revised version.
> > I will raise my ratings to 6.
> > Thank you.

---

### Official Review · Reviewer_bsUa · 2025-11-04

**Soundness:** 3
**Presentation:** 4
**Contribution:** 3
**Rating:** 6
**Confidence:** 5

**Summary:**

The paper present DUPA, a self supervised approach for representation alignment for boosting the convergence speed of diffusion transformers. The core idea is to utilize representations from a parallel branch which has the input of the same image but at a different noise level Such a mode of training would force the model to learn noise robust features faster leading to faster convergence. Experiments show an improvement of training speed by 5x and the model achieving similar performance to REPA.

**Strengths:**

1. The idea of utilizing augmented versions of the same image as input and aligning their representations is a clever approach for representation alignment. Such a mode of training may scale better for text to image training when compared to a vision encoder that introduces an inductive bias
2. The idea is novel and solid and the authors have performed extensive experiments to find the suitable layers and hyperparameters for DUPA.
3. The paper is well written and easy to follow.

**Weaknesses:**

1. Is there some contraints on the value of the independently sampled timesteps needed for better performance? As an example, assume that one timestep is sampled with maximum noise and the other at minimum noise, aligning their representations might be a case where training with DUPA loss might not leading to a meaningful solution.
2. Would performing DUPA on multiple layers at the same time lead to a better performance?
3. I think the claim regarding 10x inference speed may be a bit misleading. Usually diffusion models are utilized to obtain a few samples. I believe with the current setup, DUPA will portray similar sampling speeds to REPA. I would advise the authors to correct this wording.

**Questions:**

1. Aside from the cosine similarity loss similar to REPA, would a simple MSE loss work for DUPA? In the case of REPA the cosine similarity loss may make sense. But is it the same case here, since the features of the same network are aligned?
2. Could the authors provide a comparison between REPA and DUPA for text to image generation ?
3. I'm rating the paper as borderline accept now, mainly because this approach seems scalable for text to image generation on the first look. ,but I'm willing to improve my rating if the authors can address the questions in a satisfactory way.

---

> ### Author Response · Authors · 2025-11-21
> **Response to Reviewer bsUa (1/2)**
>
> We sincerely appreciate your valuable time and effort spent reviewing our paper. We will correct the inappropriate expressions you pointed out in the revised version Our response to your question is as follows:
>
> $\textbf{[Q1]}$ Is there some contraints on the value of the independently sampled timesteps needed for better performance? As an example, assume that one timestep is sampled with maximum noise and the other at minimum noise, aligning their representations might be a case where training with DUPA loss might not leading to a meaningful solution.
>
> $\textbf{[A1]}$ In similar self-alignment works (e.g., SRA[1], SD-DiT[2]), it is generally believed that using t=$\sigma_{min}$ in the teacher branch (typically frozen) is optimal.
>
> As for DUPA, to investigate the impact of timestep selection, we conduct experiments under three configurations: using only dual-path sampling, using only condition alignment, and using both improvements simultaneously. For experimental efficiency, we conduct tests on ImageNet at 256×256 resolution using DUPA-B/2, training for 80 epochs without using CFG. Time intervals in the table below denote the range from which t is sampled for one branch of dual-path sampling, while the other branch retains the original sampling strategy. We employ uniform sampling across the time interval.
>
> 1. Only dual-path sampling.
>
> | Time Interval | FID-50K |
> | ------------- | ------- |
> | \[0,0.1)      | 28.92   |
> | \[0,0.2)      | 28.14   |
> | \[0,0.3)      | 27.45   |
> | \[0,1)        | 26.21   |
>
> A broader sampling range enables the model to encounter more diverse intermediate states $z_t$, thereby enhancing performance.
>
> 2. Only condition alignment.
>
> To investigate the impact of timestep selection on condition alignment, we apply the stop-gradient operation to one branch of the dual-path sampling (which can be regarded as the teacher branch), utilizing only the intermediate conditions output by the teacher branch for condition alignment without computing the diffusion loss of the teacher branch.
>
> | Time Interval | FID-50K |
> | ------------- | ------- |
> | \[0,0.1)      | 27.21   |
> | \[0,0.2)      | 27.04   |
> | \[0,0.3)      | 27.13   |
> | \[0,1)        | 28.17   |
> | \[0.8,1)      | 30.36   |
>
>
> Selecting a relatively small t (closer to the clean image) in the teacher branch is most beneficial for model performance. This is understandable because when the teacher branch is frozen, its output effectively serves as "ground truth" that guides the model. Inaccurate outputs generated from large t (blurred images) would harm model performance.
>
> 3. Both dual-path sampling and condition alignment.
>
> | Time Interval | FID-50K |
> | ------------- | ------- |
> | \[0,0.1)      | 26.17   |
> | \[0,0.2)      | 25.72   |
> | \[0,0.3)      | 25.58   |
> | \[0,1)        | 25.23   |
>
> When both improvements are adopted simultaneously, the result of not restricting the selection of t is better, which also reflects the simplicity of the proposed method as it does not require too much manual configuration.
>
> Meanwhile, to address your concern that a large discrepancy between t1 and t2 might be meaningless, we constrained the gap range between them (specifically, by first sampling t1 normally, then uniformly sampling t2 within a specified range). The results are as follows.
>
> | \|t₁−t₂\| | FID-50K |
> |-------------|---------|
> | \[0,0.2)     | 26.31   |
> | \[0,0.4)     | 25.48   |
> | \[0,0.6)     | 25.12   |
> | \[0,0.8)     | 25.20   |
> | \[0,1)       | 25.23   |
>
> The results show that a gap range that is too small reduces performance due to decreased diversity in the training data. When the gap range is unrestricted, there is a slight performance drop (likely because only a few t pairs approach the maximum gap), but considering the small impact, we believe that not restricting the t range is a more straightforward and transferable approach.
>
> [1] Jiang, Dengyang, et al. "No Other Representation Component Is Needed: Diffusion Transformers Can Provide Representation Guidance by Themselves." arXiv preprint arXiv:2505.02831 (2025).
>
> [2] Zhu, Rui, et al. "SD-DiT: Unleashing the power of self-supervised discrimination in diffusion transformer." Proceedings of the IEEE/CVF Conference on Computer Vision and Pattern Recognition. 2024.

---

> ### Author Response · Authors · 2025-11-21
> **Response to Reviewer bsUa (2/2)**
>
> $\textbf{[Q2]}$  Would performing DUPA on multiple layers at the same time lead to a better performance?
>
> $\textbf{[A2]}$ During early exploratory experiments using SiT as the base model, we found that simultaneously aligning the early layers indeed improved performance. However, when using DDT as the base model, aligning the output layer of the decoupled condition encoder achieved optimal results, surpassing the performance obtained by simultaneously aligning early layers in SiT.
>
> $\textbf{[Q3]}$ Aside from the cosine similarity loss similar to REPA, would a simple MSE loss work for DUPA? In the case of REPA the cosine similarity loss may make sense. But is it the same case here, since the features of the same network are aligned?
>
> $\textbf{[A3]}$ Thanks for your advice. We conduct experiments using MSE loss, with results shown below. For experimental efficiency, we train DUPA-B/2 on ImageNet at 256×256 resolution for 80 epochs without using CFG.
>
> | Alignment Loss | FID-50K |
> | -------------- | ------- |
> | Cos. sim.      | 25.23   |
> | MSE            | 26.15   |
>
> We hypothesize that the slightly inferior performance of MSE loss is due to directly aligning conditions across different timesteps t, which slightly degrades model performance. Therefore, we keep t identical on both branches in dual-path sampling; results are shown below.
>
> | Alignment Loss | FID-50K |
> | -------------- | ------- |
> | Cos. sim.      | 27.86   |
> | MSE            | 27.83  |
>
>
> The results demonstrate that cosine similarity and MSE loss now yield consistent performance.
>
> $\textbf{[Q4]}$ Could the authors provide a comparison between REPA and DUPA for text to image generation?
>
> $\textbf{[A4]}$ Thanks for your instructive suggestions, we are currently conducting text-to-image experiments on MMDiT[1] and will provide timely updates if progress is made.
>
> [1] Patrick Esser, et al. "Scaling Rectified Flow Transformers for High-Resolution Image Synthesis." ICML 2024

---

> > ### Author Response · Authors · 2025-12-03
> > **Response to Reviewer bsUa**
> >
> > $\textbf{[Q4]}$ Could the authors provide a comparison between REPA and DUPA for text to image generation?
> >
> > $\textbf{[A4]}$ For a more accurate comparison, we use the MMDiT[1] implementation from the REPA codebase and conduct experiments on MS-COCO[2] under the same settings. The results are as follows.
> >
> > | Method | FID |
> > |--------|-----|
> > | MMDiT (SDE;NFE=250,150Kiter) | 5.30 |
> > | MMDiT+DUPA (SDE; NFE=250,150Kiter)  | 4.37 |
> > | MMDiT+REPA (SDE; NFE=250,150Kiter)  | 4.14 |
> >
> > From the table results, we can see that even with only internal self-supervision, DUPA can still achieve performance close to REPA on t2i task.
> >
> > [1] Patrick Esser, et al. "Scaling Rectified Flow Transformers for High-Resolution Image Synthesis." ICML 2024
> > [2] Microsoft COCO: Common Objects in Context, ECCV 2014

---

### Official Review · Reviewer_tRWZ · 2025-11-04

**Soundness:** 3
**Presentation:** 1
**Contribution:** 2
**Rating:** 6
**Confidence:** 3

**Summary:**

The paper proposes DUPA (Dual-Path Condition Alignment), an unsupervised representation-alignment framework for training diffusion transformers. Building upon the decoupled diffusion transformer (DDT), DUPA introduces a condition alignment loss that aligns features of multiple noisy versions of the same image, effectively mimicking the self-supervised contrastive learning.
Experimental results on ImageNet 256×256 demonstrate that DUPA outperforms the reproduced DDT baseline both in terms of training and sampling efficiency.

**Strengths:**

- The proposed DUPA not only achieves better FID than DDT (reproduced) with the same number of training steps but also reduces the number of denoising steps for sampling.
- The design of DUPA is simple and easy to integrate.

**Weaknesses:**

- The experiments are limited to ImageNet 256×256.
- While the paper is clear in structure, the prose is dense and overly formal in places, which introduces unnecessary friction. For example, in lines 192–194:
	- there are too many dependent clauses. "thereby generating... to be denoised".
	- Unnecessary formality. "conduct multiple samplings to get different..." is verbose. "We sample multiple noises.." will be more natural.
	- Grammar issues: "independent sampling times" -> "independent samples"

**Questions:**

- What's the main difference between DDT and the dual-path sampling in Table 4? Aren't they the same without DUPAlign loss?
- Why does DUPA not integrate the architectural improvements of DDT? Would DUPA still retain its advantage if those were included?
- How does DUPA work on higher resolution like ImageNet512?

---

> ### Author Response · Authors · 2025-11-21
> **Response to Reviewer tRWZ**
>
> We sincerely appreciate your valuable time and effort spent reviewing our paper. We will correct the inappropriate expressions you pointed out in the revised version. Our response to your question is as follows:
>
> $\textbf{[Q1]}$ What's the main difference between DDT and the dual-path sampling in Table 4? Aren't they the same without DUPAlign loss?
>
> $\textbf{[A1]}$ DDT with dual-path sampling will predict velocity and calculate diffusion loss separately for two independent noisings, and then add two diffusion losses together as the total diffusion loss, which proves more efficient than increasing batch size or training iterations.
>
> $\textbf{[Q2]}$ Why does DUPA not integrate the architectural improvements of DDT? Would DUPA still retain its advantage if those were included?
>
> $\textbf{[A2]}$ In practice, we first change the structure of SiT to DDT before conducting experiments, primarily for the following reasons:
>
> 1. Many recent works (e.g., REPA[1], $\Delta$FM[2], Dispersive Loss[3]) leveraging representation learning to assist generative model training—including REPA—have been implemented on SiT codebase. Doing so allows us to exclude irrelevant factors (e.g., hyperparameter settings, architectural improvements, sampling methods) for fair comparison.
>
> 2. Our method only utilizes DDT's core improvement (decoupling DiT's early layers as a condition encoder), which is straightforward to implement within SiT. Introducing  architectural improvements unrelated to our innovation would require additional ablation experiments.
>
> Since DDT's architectural improvements are engineering-based whereas our method is principled and architecture-agnostic, incorporating these optimizations should theoretically yield further performance gains. Due to time limits, we currently lack results for this configuration but will provide timely updates if progress is made.
>
> $\textbf{[Q3]}$ How does DUPA work on higher resolution like ImageNet512?
>
> $\textbf{[A3]}$ Results on ImageNet 512x512 are presented below. Each model is trained for 80 epochs, and the results of REPA are taken from the original paper.
>
> | Model     | CFG   | FID↓  | sFID↓ | IS↑    | Pre.↑ | Rec.↑ |
> | --------- | ----- | ---- | ---- | ----- | ---- | ---- |
> | SiT-B/2   | ✗   | 34.3 | 6.69 | 47.0  | 0.59 | 0.63 |
> | DDT-B/2   | ✗   | 29.1 | 6.42 | 53.8  | 0.62 | 0.64 |
> | DUPA-B/2  | ✗ | 26.8 | 6.17 | 62.1  | 0.66 | 0.64 |
> | SiT-XL/2  | ✗  | 19.2 | 5.37 | 86.1  | 0.67 | 0.62 |
> | DDT-XL/2  | ✗ | 13.3 | 4.88 | 102.9 | 0.69 | 0.62 |
> | DUPA-XL/2 | ✗ | 9.14 | 4.70 | 125.8 | 0.72 | 0.63 |
> | DUPA-XL/2 | ✓ | 2.83 | 4.53 | 229.6 | 0.83 | 0.59 |
> | REPA-XL/2 | ✓ | 2.44 | 4.21 | 247.3 | 0.84 | 0.56 |
>
>
> [1]Sihyun Yu, et al. "Representation Alignment for Generation: Training Diffusion Transformers Is Easier Than You Think." ICLR 2025
>
> [2]George Stoica, et al. "Contrastive Flow Matching." ICCV 2025
>
> [3]Wang Runqian, et al. "Diffuse and Disperse: Image Generation with Representation Regularization."  arXiv preprint arXiv:2506.09027 (2025)

---

### Author Response · Authors · 2025-12-03
**General Response**

We have uploaded a revised version, which includes the following modifications:

1. Revised redundant statements in lines 192~194. (to reviewer tRWZ)
2. Revised the wording regarding accelerated sampling. (to reviewer bsUa)
3. Added experiments on timestamps in the appendix. (to reviewer 7oUe)

For the convenience of comparison, the background of the modified part is light blue. Thanks for your meticulous review and suggestions of our paper.

---

### Author Response · Authors · 2025-12-04
**Summary of the Score Changes and Discussions in Rebuttal**

Dear Area Chair,

For the convenience of your review, we have made the following summary for rebuttal.

We propose DUPA, a method that aligns intermediate representations across different denoising paths for the same image, which could accelerate generative model training and inference.

Before the system freezed, our submission had effectively **moved from a mixed state (6, 6, 4, 6) to a positive state (6, 6, 6, 8)**, with two reviewers (**7oUe and A859**) explicitly confirming score increases.

Here we summarize the score changes and the discussion during rebuttal.

1. Reviewer tRWZ raised questions regarding the design of DUPA and its performance on high-resolution datasets. We have provided detailed responses to these concerns and supplemented additional results on ImageNet 512x512.

2. Reviewer bsUa inquired about timestamp selection and loss design, and suggested conducting experiments on text-to-image (T2I) tasks. We performed ablation experiments on timestamps and loss functions, and have included new results on T2I task. Although bsUa does not have a chance to reply, bsUa explicitly stated in the initial comments: **I'm willing to improve my rating if the authors can address the questions in a satisfactory way**.

3. Reviewer 7oUe asked questions concerning timestamps and comparisons with related work. We have supplemented the relevant experiments and provided detailed comparisons in methodology and results with the works listed by 7oUe. Reviewer 7oUe increased score to 6 and responded with **Thanks for your replyment and sufficient exps from issues, which solved my questions. Please these addtional exps/analysis in your revised version. I will raise my ratings to 6. Thank you**.

4. Reviewer A859 queried about results on domain-specific datasets and certain experimental details. We addressed these by citing the original text and conducting experiments on the medical dataset ChestX-ray 8. Reviewer A859 raised score to 8 and replied **Thank you for the rebuttal. The replies have addressed my concerns. Thus, I will raise the rating to 8 and recommend acceptance**.

We sincerely thank you for the additional time and effort devoted to reviewing and evaluating our submission and hope these clearly comments and score changes will be taken into consideration when making the final recommendation.

Best,

Authors

---

### Meta-Review · Area_Chair_z58h · 2026-01-08

**Summary:**

Reviewers mostly liked the idea and results but felt the original evidence and framing were a bit thin. The biggest worry was limited evaluation (mostly IN256), so it was hard to trust claims about scaling, high-res, and "real-world" domains. They also wanted clearer differentiation from nearby self-alignment papers (SD-DiT, SRA, etc.) and a better explanation for why the alignment should help convergence. Finally, there were practical design questions (timestep choices, loss choice, multi-layer alignment) plus some wording/presentation issues and skepticism about the 10x inference speed framing. Most concerns were convincingly addressed in the rebuttal, and considering the reviewers comments and discussions, the AC recommends acceptance.

**Reviewer Concerns:**

The rebuttal handled most major issues: it added IN 512x512 results, a domain-specific dataset (ChestX), and a text-to-image comparison (MS-COCO on MMDiT), which directly improved the generality story. It also added detailed ablations on timestep ranges, and cosine vs MSE loss, plus discussion and comparisons vs SD-DiT/SRA/others. What’s still not fully resolved is broader "OOD vs strong encoders" evidence beyond a couple domains, and the missing experiment that combines DUPA with all DDT engineering tweaks (authors argue it should help, but didn’t show it).

**Reviewer Scores:**

tRWZ: likely stays at 6 since high-res results and clarifications address the main asks, but they sounded cautious overall.
bsUa: likely stays at 6 because the rebuttal hits their timestep/loss questions, fixes the inference wording, and adds the T2I comparison they requested, and they explicitly said they’d upgrade if satisfied.
7oUe: confirmed increasing the rating in-thread after seeing the added discussion/experiments.
A859: confirmed increase after ChestX-ray8 and the batch-size vs sampling-times clarification.

---

### Decision · Program_Chairs · 2026-01-26

Accept (Poster)